# Molecular detection of *Coxiella burnetii* infection in aborted samples of domestic ruminants in Iran

**Ashraf Mohabati Mobarez[1], Mohammad Khalili[2], Ehsan Mostafavi[3,4], Saber Esmaeili [3,4]***

**1** Department of Bacteriology, Faculty of Medical Sciences, Tarbiat Modares University, Tehran, Iran,
**2** Department of Pathobiology, Faculty of Veterinary Medicine, Shahid Bahonar University of Kerman, Kerman, Iran, **3** Department of Epidemiology and Biostatics, Research Centre for Emerging and Reemerging Infectious Diseases, Pasteur Institute of Iran, Tehran, Iran, **4** National Reference Laboratory of Plague, Tularemia and Q Fever, Research Centre for Emerging and Reemerging Infectious Diseases, Pasteur Institute of Iran, Akanlu, Kabudar-Ahang, Hamadan, Iran

* dr.saberesmaeili@gmail.com, s-esmaeili@pasteur.ac.ir

**Data Availability Statement:** All relevant data are within the paper and its Supporting information files.

**Funding:** This work was supported financially by Tarbiat Modares University (Tehran, Iran), Pasteur

## Abstract

### Background

*Coxiella burnetii* is the causative agent of Q fever which is a highly infectious zoonotic disease. *C. burnetii* has become one of the most important causes of abortion in livestock, which can lead to widespread abortions in these animals. There are very limited studies on the prevalence of *C. burnetii* infection in cases of animal abortion in Iran. The aim of this study was to investigate the occurrence of *C. burnetii* in ruminant abortion samples in Iran.

### Methods

Abortion samples from cattle, sheep and goats were collected from different parts of Iran and were tested using Real-time PCR targeting the IS1111 element of *C. burnetii*.

### Results

In this study, 36 samples (24.7%) of the 146 collected samples were positive for *C. burnetii*. The prevalence of *C. burnetii* was 21.3% (20 of 94 samples) in sheep samples. Also, 10 of 46 cattle samples (21.7%) were positive. All six goat abortion samples were positive for *C. burnetii*.

### Conclusions

The findings of the study demonstrate that *C. burnetii* plays an important role in domestic ruminant abortions in Iran, suggesting that more attention should be paid to the role of *C. burnetii* in domestic animal abortions by veterinary organizations. The risk of transmitting the infection to humans due to abortion of animals should also be considered.

Institute of Iran and Centre for Communicable Diseases Control in Ministry of Health (grant 810), and also, Iranian National Scientific Foundation (INSF; Contracted No.91004716). The funders not had any role in study design, conducting survey and results analysis.

**Competing interests:** The authors have declared that no competing interests exist.

## Background

*Coxiella burnetii* is a small, Gram-negative coccobacillus and obligate intracellular bacterium. *C. burnetii* is the causative agent of Q fever which is a highly infectious zoonotic disease. Based on 16S rRNA sequence analysis, *C. burnetii* belongs to the Gama-subdivision of Proteobacteria within the *Legionellales* order and family *Coxiellaceae* [1].

Domestic ruminants are the main reservoirs of *C. burnetii*. Q fever in cattle, sheep, and goats is usually asymptomatic but in some cases is associated with pneumonia and reproductive disorders, including abortion, stillbirth, endometritis, and infertility [2]. *C. burnetii* is excreted by infected animals into milk, feces, urine, placenta, fetal fluids, and vaginal discharge. The main route of transmission to humans is through inhalation of aerosols and dust particles contaminated with *C. burnetii* [1, 2]. People at risk of the disease include farmers, veterinarians, butchers, slaughterhouse workers, farmers, and people in contact with domestic animals', especially during farm animals'delivery [3]. Easy and fast release into aerosols, survival in extreme environmental conditions as well as a low infectious dose of the bacterium have made *C. burnetii* a very serious biological threat to military personnel and civilians [4, 5].

Q fever manifests in two forms including acute and chronic disease in humans. Acute Q fever is mainly a self-limiting, febrile illness with flu-like disease and is asymptomatic in 60% of cases [3]. Persistence of *C. burnetii* infection in humans (in acute and asymptomatic cases) can lead to the chronic form of Q fever. Endocarditis is the main clinical manifestation of chronic Q fever and leads to death in over 65% of the patients, if untreated [1, 6].

In recent years, *C. burnetii* has become one of the most important causes of abortion in domestic ruminants in all countries, which can lead to widespread abortions in animals [7]. Although Q fever can affect many domestic animals, the effects on abortion are more likely to appear in small ruminants. Also, abortion and the accompanying excretory fluids are one of the main routes of contaminating the environment, which can lead to widespread infection among animals and human populations [6]. There is a very high density of *C. burnetii* in amniotic fluid and the placenta during delivery in infected animals. Aborted products can be easily aerosolized and transmitted by wind up to several kilometers from the site of the abortion [8]. Therefore, the role of *C. burnetii* in cases of animal abortion is of great importance both in terms of epidemiology and in the transmission risk to humans.

In recent years, seroepidemiological studies in animal and human populations in Iran have shown that Q fever is an endemic disease [9]. However, clinical cases of Q fever are rarely diagnosed and reported by the health system in Iran [9]. On the other hand, there are very limited studies on the prevalence of *C. burnetii* in cases of animal abortion in Iran. Also, conducting various studies and showing the importance of the disease can increase awareness in the health system and clinical physicians to Q fever in Iran. The aim of this study was to conduct a molecular investigation of *C. burnetii* in domestic ruminant abortion samples in Iran.

## Methods

### Sample collection

This study was conducted during 2017–2018 with the cooperation of clinical veterinarians as well as the Iranian Veterinary Organization. Domestic ruminant aborted samples were collected from different provinces of Iran and samples included spleen and liver of aborted fetuses, and abortion fluids and placenta cotyledons of aborted animals (cattle, sheep and goat). Sampling was performed in accordance with biological safety rules. Samples were frozen immediately after obtaining and were transported to the laboratory by maintaining the cold chain. Laboratory work was done under biosafety level 2 (BSL2) laboratory settings.

This study was approved by the Ethics Committee for Biomedical Research of Tarbiat Modares University (Ethic Code: IR.TMU.REC.1395.510). The Ethics Committee for Biomedical Research of Tarbiat Modares University approved the consent procedure, the proposal and protocol of this study.

### DNA extraction

Genomic DNA was isolated using the Roche High Pure PCR Template Preparation Kit (Roche, Germany), according to the manufacturer's instructions. For tissue samples, 50 mg from each sample was used for DNA extraction. Also, 200 µL of abortion fluids was used for DNA extraction. All extracted DNA was stored at -20 ˚C until molecular testing.

### Real-time Polymerase Chain Reaction

Real-time PCR was performed using specific primers and probe sequences targeting the IS1111 element of *C. burnetii*. Real-time PCR reactions were performed using the following reaction mixture: 10 µL of 2x RealQ Plus Master Mix for Probe (Ampliqon, Denmark), 900 nM forward primer (AAAACGGATAAAAAGAGTCTGTGGTT), 900 nM reverse primer (CCACACAAGCGCGATTCAT), 200 nM probe (6-FAM-AAAGCACTCATTGAGCGCCGCG-TAMRA) and 4 µL of DNA template [10]. Real-time PCR was performed on the Corbett 6000 Rotor-Gene system (Corbett, Victoria, Australia), with a final volume of 20 µL for each reaction. The PCR amplification program was 10 minutes at 95˚C, followed by 45 cycles of 15 s at 94˚C and 60 s at 60˚C. DNA of the Nine Mile strain (RSA 493), was used as a positive control and double distilled water was used as a negative control. Results were generated using Rotor-Gene® Q 2.3.5 software (QIAGEN). Samples showing cycle threshold (Ct) values of 36 or lower for *C. burnetii* IS1111 qPCR assays were considered positive [10]. The positive samples were tested three times by RT-PCR. Also, PCR products of positive samples were visualized by 2% agarose gel electrophoresis (70 bp). Finally, the samples that had a first positive test and the next two tests were positive and also had the acceptable amplicon size in the agarose gel electrophoresis were considered as true positive.

## Results

A total of 146 samples of abortions were collected from different parts of Iran, of which 94 samples (specimens included 17 spleens and 24 livers of aborted-fetuses, 33 abortion fluids and 45 cotyledons of aborted animals) were from sheep abortions (S1 Table), 46 samples (specimens included 40 spleens and 12 livers of aborted-fetuses, and 5 cotyledons of aborted animals) were from cattle abortions (S2 Table), and 6 samples were from goat abortions (specimens included 3 spleens of aborted-fetuses and 5 abortion fluids).

In total, 36 samples (24.7%, 95% Confidence Interval: 18.1–32.6) of the 146 collected samples were positive for *C. burnetii* using Real-time PCR. The prevalence of *C. burnetii* was 21.3% (95% CI: 13.8–31.2) (20 of 94 samples) in aborted sheep samples. The highest prevalence of *C. burnetii* was detected in Tehran (54.5%, 95% CI: 24.6–81.9), Mazandaran (50%, 95% CI: 9.2–90.8), West-Azarbaijan (40%, 95% CI: 7.3–83.0) and Hamadan (38.5%, 95% CI: 15.1–67.7) provinces, respectively (Table 1).

Based on results, 10 of 46 cattle-aborted samples (21.7%, 95% CI: 11.4–36.8) were positive for *C. burnetii*. The highest prevalence of *C. burnetii* was detected in Tehran (24.3%, 95% CI: 12.7–41.5), and Alborz (16.7%, 95% CI: 0.9–63.5) provinces, respectively (Table 1).

All six goat-aborted samples were positive for *C. burnetii*. Two and four of the samples were collected from West-Azarbaijan and Tehran provinces, respectively.

**Table 1. The prevalence of *C. burnetii* in sheep and cattle abortion samples by Real-time PCR in Iran during 2017–2018.**

| Province | Sheep Samples | | Cattle samples | |
| --- | --- | --- | --- | --- |
| | Samples (N) | No of Positive Samples (%, 95% CI*) | Samples (N) | No of Positive Samples (%, 95% CI*) |
| Tehran | 11 | 6 (54.5, 24.6–81.9) | 37 | 9 (24.3, 12.7–41.5) |
| West-Azarbaijan | 5 | 2 (40.0, 7.3–83.0) | - | - |
| East-Azarbaijan | - | - | 1 | 0 (0.0, 0.0–94.5) |
| Ardabil | 34 | 2 (5.9, 1.0–21.0) | 1 | 0 (0.0, 0.0–94.5) |
| North-Khorasan | 20 | 3 (15.0, 4.0–38.9) | - | - |
| Razavi-Khorasan | 7 | 0 (0.0, 0.0–43.9) | - | - |
| Hamadan | 13 | 5 (38.5, 15.1–67.7) | 1 | 0 (0.0, 0.0–94.5) |
| Alborz | - | - | 6 | 1 (16.7, 0.9–63.5) |
| Mazandaran | 4 | 2 (50.0, 9.2–90.8) | - | - |
| Total | 94 | 20 (21.3, 13.8–31.2) | 46 | 10 (21.7, 11.4–36.8) |

* 95% Confidence Interval

## Discussion

The present study was conducted to investigate the prevalence of *C. burnetii* in ruminant abortions in Iran and showed that 24.7% of the collected samples were positive for *C. burnetii* using Real-time PCR. Based on the results of this study, it is recommended that the involvement of *C. burnetii* be seriously considered in cases of animal abortion and the possibility of transmission to humans regarded. More than 50 years have passed since the first report of Q fever, yet it is still a neglected disease in Iran. Moreover, despite the fact that this disease has a very significant seroprevalence among livestock and human populations, there is no system for registering and reporting or a program for diagnosis and prevention in the human health care system and the veterinary organization in Iran [9]. Therefore, conducting more studies on suspected patients and conducting molecular epidemiological research will shed light on the epidemiological situation of Q fever in Iran and will encourage more attention to the human and animal health system towards this disease. Actually Q fever is one of the causes of endocarditis in humans, and it was reported that about 31% of culture-negative endocarditis in Iran is caused by *C. burnetii* [11].

In the current study, 21.3% of sheep-aborted samples were positive for *C. burnetii*. Tehran (54.5%), Mazandaran (50%), West-Azarbaijan (40%) and Hamadan (38.5%) provinces had the highest frequency of *C. burnetii* positivity in sheep-aborted samples, respectively. Prevalence in our study was higher than other similar studies in Iran. Molecular prevalence of *C. burnetii* in sheep abortions in Mashhad county (northeastern Iran), Sistan and Baluchestan province (southeastern Iran) and Fars province (southern Iran) were reported as 17.3%, 16.6% and 2.7%, respectively [12–14]. All of these studies used convenience sampling similar to our study. One of the reasons for the high prevalence in our study was the use of a much more sensitive method (Real-time PCR) for detection of *C. burnetii*. Other reasons include different study areas and increased prevalence over time. In other countries, different rates were reported; 33.6% in Egypt [15], 2–11% in Turkey [16, 17], 21.5% in Italy [18], 47.6% in Hungary [19] and 44.4% in Switzerland [20]. Based on a recent study in Iran, shedding of *C. burnetii* into milk was high (35.7%) among sheep with an abortion history [21]. Therefore, it seems that abortion in sheep due to *C. burnetii* should be given more attention, because in addition to the huge economic damage caused by ruminant abortions, a large amount of *C. burnetii* is released into the environment by abortion products and vaginal discharges which can infected human and animals.

In our study, *C. burnetii* was detected in 21.7% of cattle-aborted samples. Also, Tehran (24.3%), and Alborz (16.7%) provinces had the highest prevalence of *C. burnetii* in cattle-aborted samples. However, the very small number of samples from other provinces made it difficult to compare in our study. In a study in Mashhad county, 25% of cattle fetuses were positive for *C. burnetii* [12]. In other countries, the prevalence of *C. burnetii* in cattle abortions was 4% in Turkey [17], 11.6% in Italy [18], 25.9% in Hungary [19] and 35% in Cyprus [22]. Also, *C. burnetii* was detected in 33.3% of milk samples of cattle with an abortion history in Iran [21]. According to the findings of this study and other similar studies in Iran and other countries, it is recommended that control measures be taken against this bacterium in order to prevent abortions and Q fever in Iran.

One of the limitations of this study was the small number of samples, which made it difficult to make prevalence comparisons. Therefore, it is suggested that a greater number of samples be collected from different farms in future studies. It is also recommended that non-convenience sampling be used in future studies. Unfortunately, one of the limitations of our work was the failure to record the age of the aborted fetuses. Having information on the age of aborted fetuses, can inform the epidemiology of *C. burnetii* abortions in domestic ruminants.

## Conclusions

The findings of the study demonstrate that *C. burnetii* plays an important role in domestic ruminant abortions in Iran and more attention should be paid to Q fever by the health care system and veterinary organization. Also, necessary training in the prevention of this disease should be provided to ranchers and at-risk people.

## Supporting information

**S1 Table. Results of sheep abortion sampling.**
(DOCX)

**S2 Table. Results of cow abortion sampling.**
(DOCX)

## Acknowledgments

We would like to express our gratitude to the large-animal veterinarians for their help in sampling.

## Author Contributions

**Conceptualization:** Ashraf Mohabati Mobarez, Mohammad Khalili, Ehsan Mostafavi, Saber Esmaeili.

**Data curation:** Saber Esmaeili.

**Formal analysis:** Ehsan Mostafavi, Saber Esmaeili.

**Funding acquisition:** Ashraf Mohabati Mobarez, Saber Esmaeili.

**Investigation:** Saber Esmaeili.

**Methodology:** Ashraf Mohabati Mobarez, Mohammad Khalili, Ehsan Mostafavi, Saber Esmaeili.

**Project administration:** Saber Esmaeili.

**Resources:** Saber Esmaeili.

**Software:** Saber Esmaeili.

**Supervision:** Ashraf Mohabati Mobarez, Mohammad Khalili, Ehsan Mostafavi.

**Validation:** Mohammad Khalili, Saber Esmaeili.

**Visualization:** Saber Esmaeili.

**Writing – original draft:** Ehsan Mostafavi, Saber Esmaeili.

**Writing – review & editing:** Ashraf Mohabati Mobarez, Mohammad Khalili, Ehsan Mostafavi, Saber Esmaeili.

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
