## [Decision Letter · Decision Letter 0]

17 Dec 2020

PONE-D-20-32185

Molecular detection of Coxiella burnetii infection in aborted samples of livestock in Iran

PLOS ONE

Dear Dr. Esmaeili,

Thank you for submitting your manuscript to PLOS ONE. After careful consideration, we feel that it has merit but does not fully meet PLOS ONE’s publication criteria as it currently stands. Therefore, we invite you to submit a revised version of the manuscript that addresses the points raised during the review process.

Please attend to all the comments and concerns raised by the reviewers.

The major drawback of this work is that convenient sampling was used, which makes generalisation 

of the results beyond this study invalid. In addition, the samples collected from some areas, especially for cattle was too small, making the estimates less meaningful. Therefore, the authors should account for these limitations of this study in their discussion and take them into consideration as they compare their results to those of others.

We look forward to receiving your revised manuscript.

Kind regards,

Martin Chtolongo Simuunza, PhD

Academic Editor

PLOS ONE

Journal Requirements:

3. Please provide a list of the specific farms that livestock abortions were obtained from.

Reviewers' comments:

Reviewer's Responses to Questions

**Comments to the Author**

1. Is the manuscript technically sound, and do the data support the conclusions?

Reviewer #1: Yes

Reviewer #2: Partly

2. Has the statistical analysis been performed appropriately and rigorously? 

Reviewer #1: No

Reviewer #2: N/A

3. Have the authors made all data underlying the findings in their manuscript fully available?

Reviewer #1: Yes

Reviewer #2: Yes

4. Is the manuscript presented in an intelligible fashion and written in standard English?

Reviewer #1: Yes

Reviewer #2: No

5. Review Comments to the Author

Reviewer #1: The manuscript by Mobarez et al., describes a survey of 146 tissue samples from spontaneously-aborted cattle, sheep, and goats for the presence of Coxiella burnetii by RT-PCR, using IS1111 as a target. Based on previous reports, results of the study are not surprising. Moreover, originality of the work is nominal and rests solely on the fact that Iranian livestock abortion samples were tested.

Issues that need to be addressed:

1. There are numerous English grammar and syntax errors in the text. My suggestions to the authors are itemized below.

2. How did the authors establish that ≤ 36 cycles was the RT-PCR Ct value for positivity?

3. How many RT-PCR analyses were done per sample? Were any technical replicates performed? Did the authors check any PCR products on agarose gels to ensure that the amplicon size was correct for the target?

4. The Discussion section tends to be redundant with the Results section, and repeats much of the same information.

5. Please use standard journal abbreviations in the References section.

Suggestions to improve English grammar:

Line 13- should read “Coxiella burnetii is the causative…is a highly…”

Line 18- replace “goat” with “goats”

Lines 19 and 89- should read “…targeting the IS1111 element of…” (This is not a gene, but rather a transposable element).

Line 24- should read “…C. burnetii plays an important…”

Line 25- should read “…Iran, suggesting that more…”

Line 30- should read “…small, Gram-negative coccobacillus and…”

Line 31- should read “…is the causative…is a highly…”

Line 32- should read “…16S rRNA… belongs to the gamma…” (separate 16S and rRNA, belongs in place of belonged and misspelled gamma).

Line 34- should read “…fever manifests in two forms including acute and chronic disease in humans.”

Line 38- should read “…of chronic Q fever and leads…”

Line 44- replace “through” with “into”

Lines 45-46- condense the list of high-risk groups.

Lines 47-50- This is an awkward sentence. I would suggest, “Easy and fast release into aerosols, survival in harsh environmental conditions as well as a low infectious dose of the ….”

Lines 49-50- It is unclear why “military personnel” is included here. Is this due to its prior use in biological weapons? It looks odd without a brief explanation.

Line 51- replace “livestock’s” with “livestock”

Line 53- should read “…main routes of transmission in….”

Line 55- should read “…during birth in…”

Line 60- delete “in Iran”, as it is redundant with prior line

Line 63- should read “…can increase awareness in the health…”

Line 64- should read “…was to conduct a molecular…”

Line 70- replace “province” with “provinces”

Line 73- “conditions” should read “rules”, right?

Line 74- should read “…laboratory work was done under a BSL2 biological safety cabinet.” By the way, this level of

containment seems inadequate for C. burnetii work, especially with contaminated samples from animals.

Line 85- should read “…instructions. For tissue…”

Line 86- should read “…sample was used…”

Line 87- should read “…DNA was stored… molecular testing.”

Line 93- should read “Real-time PCR was performed…”

Line 95- should read “…program was 10…”

Line 96- should read “…of the Nine…”

Line 97- delete “with performed”

Line 105- replace “sheep’s” with “sheep”

Line 106- replace “goat’s” with “goat”

Line 111- should read “…cattle-aborted…C. burnetii.” (italicized scientific name)

Line 113- should read “…goat-aborted samples…”

Table 1 legend- delete “using”

Table 2 legend- make uniform with Table 1 legend to read “…in cattle abortion samples by Real-time PCR.”

Line 124- replace “effect” with “involvement”

Lines 126-127- should read “More than….is still a neglected disease in Iran. Moreover, despite…”

Line 128- delete “but”

Line 129- should read “…reporting or a …”

Line 132- should read “…of Q fever in Iran…”

Line 134- should read “…endocarditis in Iran is caused by C. burnetii.”

Line 135- should read “In the current study, 21.8% of sheep-aborted…positive for C. burnetii.”

Line 136- should read “…prevalence was seen in Tehran…”

Line 138- should read “…in sheep abortions…”

Line 139- should read “…south-eastern Iran)…(southern Iran)..”

Line 140- should read “…reported as 17.3%...”

Line 144- replace “Italia” with “Italy”.

Line 144- should read “Based on a recent…”

Line 145- delete “significantly” (no stats)

Line 145- should read “…sheep with an abortion…”

Line 148- should read “…cattle-aborted…”

Line 150- replace “sample s” with “samples”

Line 150- should read “…compare with our study.”

Line 152- should read “…abortions was 4%...”

Line 153- should read “…detected in 33.3% of milk…”

Line 154- should read “…with an abortion history…”

Line 155- should read “…recommended that control measures be taken against…”

Line 156- should read “…to prevent abortions and Q fever in Iran.”

Line 159- replace “have” with “plays an”

Line 174- should read “…funders had no role…”

Line 181- should read “References” (plural)

Reviewer #2: The authors present a study on ‘Molecular detection of Coxiella burnetii in aborted samples of livestock in Iran’. The objective of this study was to report the prevalence of C. burnetii in convenient samples from clinical aborted ruminants. There is novelty on the present study, considering the little information of this disease in Iran and implications for public health. Although the paper needs considerable review. I found the manuscript difficult to read and to understand, due to the sentence structure, and incorrect use of terminology. Before to address the scientific content, I advise the authors to have the manuscript reviewed for English and veterinary/ epidemiology terminology. Therefore, my comments (for this review) are general and addressing major changes regarding the scientific content.

Title :The study is focused on ruminant samples (not in livestock in general), the title should reflect that.

Abstract: The abstract used interchangeable samples from livestock, but samples are from ruminants, the reader is misled by the title and abstract (the same through out the manuscript). The study was conducted using convenience sampling submitted to the national lab (this should be described in the abstract). The conclusions are very general and open-ended. Actually, without knowing the overall prevalence in ruminants, reporting only the clinical cases/ samples submitted for diagnosis may increase the apparent prevalence.

Introduction: I found the introduction confusing, lacking of flow between the paragraphs, as the authors cite epidemiology status in animals and humans back and forward without connecting thoughts/ sentences. Also, the use of inappropriate terminology is observed in this section as for example:

Line 35: ‘chronic Q fever is extremely serious and dangerous’ in what aspect? To whom?

Line 46: ‘ dairy factory workers’ – dairy farms? Processing plants?

Line 47: ‘during livestock calving’ – delivery? Calving used for cows ( kidding for goats, etc)

Line 48: ‘ harsh environmental conditions’- do you mean extreme environmental conditions?

Line 51: ‘In recent years, C. burnetii has become one of the most important causes of abortion in livestock’s ‘ – Worldwide ? or specific country? What do you mean by ‘livestock’s’. Although Q fever can affect many domestic animals, the effects on abortion are most likely to appear in small ruminants in a naïve population. Please be specific.

Line 55 : ‘ childbirth infected animals’ – please review.

Line 57’ ‘Abortion by wind’- do you mean aborted products can be easily aerosolized and transmitted by wind? Please review.

Methods: This study uses a convenience sample from ‘cooperation of clinical veterinarians as well as the Iranian Veterinary- Organization’ – how many samples from each source? Please make a reference that this is convenience sample from clinical cases of aborted ruminants. There were different aborted specimens tested, presented how many of each type tested. Do you expect differential specificity/ sensitivity based on the sample type? If yes, what are the impacts on your findings ( add a section for limitations in the discussion section). For the PCR methods, was this protocol developed by your lab, if not please cite the protocol source.

Line 72: ’ All livestock abortions were spontaneous’. Again, here should be ruminant abortions, by spontaneous do you mean clinical? What were the age of abortion for the tested samples?

Results: The information in Tables 1 and 2 can be summarized in one table. Provide the total of positive samples and denominator for each %. e.g. line 112: 24,32%. Table legends are incomplete (add date of the study and location/ country). 95% CI should be added to the results.

Discussion: The authors made an attempt to emphasis the public health and animal health impact of the findings in aborted samples. Therefore, the need for surveillance programs to be implemented in Iran. However, one of the big pitfalls of the study is the sample size and convenience sampling, without knowing how representative are those samples of the herd prevalence, external validity, those claims are difficult to assess. One of the novel aspects of this study is that the authors used PCR comparing to most of studies used seroprevalence. How different are those studies: studied population, diagnostic testing ( Se, Sp), samples tested ( clinical versus surveillance), etc? The authors briefly described the prevalence in other studies/ countries but do not provide a justification for such numeric difference. This should be explored in the discussion section. Pitfalls are not addressed in the manuscript

6. PLOS authors have the option to publish the peer review history of their article (what does this mean?). If published, this will include your full peer review and any attached files.

Reviewer #1: No

Reviewer #2: No

---

## [Author Response · Author response to Decision Letter 0]

5 Jan 2021

Author’s Response to Reviewers Comment

Note: All changes showed in Revised Article with Changes Highlighted uploaded file and made by Track Change.

Reviewer 1#

Reviewer #1: The manuscript by Mobarez et al., describes a survey of 146 tissue samples from spontaneously-aborted cattle, sheep, and goats for the presence of Coxiella burnetii by RT-PCR, using IS1111 as a target. Based on previous reports, results of the study are not surprising. Moreover, originality of the work is nominal and rests solely on the fact that Iranian livestock abortion samples were tested.

Reviewer comment 1: There are numerous English grammar and syntax errors in the text. My suggestions to the authors are itemized below.

Author’s Response 1: Thanks for your comments. All recommended grammatical comments were considered. Manuscript was revised by a native English reviewer. 

Reviewer comment 2: How did the authors establish that ≤ 36 cycles was the RT-PCR Ct value for positivity?

Author’s Response 2: Based on our previous setting, tenfold-serial dilutions of DNA from control positive (C. burnetii Nine Mile strain) were prepared and tested by RT-PCR according to recommended reference (“Real-Time PCR with Serum Samples Is Indispensable for Early Diagnosis of Acute Q Fever, Clin Vaccine Immunol. 2010 Feb; 17(2): 286–290”). We defined the limit of detection of our assay as the lowest concentration of genome equivalents. A CT value of <36.0 was considered positive.

Reviewer comment 3: How many RT-PCR analyses were done per sample? Were any technical replicates performed? Did the authors check any PCR products on agarose gels to ensure that the amplicon size was correct for the target?

Author’s Response 3: The positive samples tested three times by RT-PCR. Also, we checked PCR products of positive samples on 2% agarose gels (70 bp). Finally, the samples that had a positive first test and the next two tests were positive and also had the acceptable amplicon size on the agarose gel were considered as true positive. 

Reviewer comment 4: The Discussion section tends to be redundant with the Results section, and repeats much of the same information.

Author’s Response 4: Thanks. Revised. 

Reviewer comment 5: Please use standard journal abbreviations in the References section.

Author’s Response 5: Done.

Reviewer 2#

Reviewer #2: The authors present a study on ‘Molecular detection of Coxiella burnetii in aborted samples of livestock in Iran’. The objective of this study was to report the prevalence of C. burnetii in convenient samples from clinical aborted ruminants. There is novelty on the present study, considering the little information of this disease in Iran and implications for public health. Although the paper needs considerable review. 

Reviewer comment 1: I found the manuscript difficult to read and to understand, due to the sentence structure, and incorrect use of terminology. Before to address the scientific content, I advise the authors to have the manuscript reviewed for English and veterinary/ epidemiology terminology. 

Author’s Response 1: Manuscript was revised by a native English reviewer.

Reviewer comment 2: Title: The study is focused on ruminant samples (not in livestock in general), the title should reflect that.

Author’s Response 2: Title was revised.

Reviewer comment 3: Abstract: The abstract used interchangeable samples from livestock, but samples are from ruminants, the reader is misled by the title and abstract (the same throughout the manuscript). The study was conducted using convenience sampling submitted to the national lab (this should be described in the abstract). The conclusions are very general and open-ended. Actually, without knowing the overall prevalence in ruminants, reporting only the clinical cases/ samples submitted for diagnosis may increase the apparent prevalence.

Author’s Response 3: Thanks. Revised. 

Abortion samples are not routinely referred and submitted to national laboratory by Veterinary organization and Clinical Veterinarians. All samples of this study were sampled only for this study. None of the samples were systematically referred to our laboratory. 

Reviewer comment 4: Introduction: I found the introduction confusing, lacking of flow between the paragraphs, as the authors cite epidemiology status in animals and humans back and forward without connecting thoughts/ sentences. Also, the use of inappropriate terminology is observed in this section as for example:

Line 35: ‘chronic Q fever is extremely serious and dangerous’ in what aspect? To whom?

Line 46: ‘dairy factory workers’ – dairy farms? Processing plants?

Line 47: ‘during livestock calving’ – delivery? Calving used for cows (kidding for goats, etc)

Line 48: ‘harsh environmental conditions’- do you mean extreme environmental conditions?

Line 51: ‘In recent years, C. burnetii has become one of the most important causes of abortion in livestock’s ‘– Worldwide? or specific country? What do you mean by ‘livestock’s’. 

Although Q fever can affect many domestic animals, the effects on abortion are most likely to appear in small ruminants in a naïve population. Please be specific.

Line 55: ‘childbirth infected animals’ – please review.

Line 57’ ‘Abortion by wind’- do you mean aborted products can be easily aerosolized and transmitted by wind? Please review.

Author’s Response 4: Thanks. The introduction section of manuscript was revised based on these comments.

Reviewer comment 5: Methods: This study uses a convenience sample from ‘cooperation of clinical veterinarians as well as the Iranian Veterinary- Organization’ – how many samples from each source? Please make a reference that this is convenience sample from clinical cases of aborted ruminants. There were different aborted specimens tested, presented how many of each type tested. Do you expect differential specificity/ sensitivity based on the sample type? If yes, what are the impacts on your findings (add a section for limitations in the discussion section). For the PCR methods, was this protocol developed by your lab, if not please cite the protocol source.

Line 72:’ All livestock abortions were spontaneous’. Again, here should be ruminant abortions, by spontaneous do you mean clinical? What were the age of abortion for the tested samples?

Author’s Response 5: The Methods section of manuscript was revised based on these comments.

Abortion samples are not routinely referred and submitted to national laboratory by Veterinary organization and Clinical Veterinarians. None of the samples were systematically referred to our laboratory. The role of the veterinary organization was to coordinate and introduce us to the farmers. All samples of this study were sampled only for this study. Finally, sampling was done by ourselves or our representative.

In most cases, only one specimen type was taken from aborted fetus samples. In cases where several specimens were taken from the aborted fetus samples, the PCR test result was completely consistent. Due to the low number of specimens ranging from an aborted fetus cannot talk properly about the sensitivity and specificity of different specimens. Specimens type was added results section.

Also, our RT-PCR test was done according to recommended reference which cited and available in PCR section of methods (“Real-Time PCR with Serum Samples Is Indispensable for Early Diagnosis of Acute Q Fever, Clin Vaccine Immunol. 2010 Feb; 17(2): 286–290”).

Unfortunately, one of the limitations of our work was the failure to record the age of the aborted fetus, which we noticed at the end of the study. This point was added to discussion section as a limitation.

Reviewer comment 6: Results: The information in Tables 1 and 2 can be summarized in one table. Provide the total of positive samples and denominator for each %. e.g. line 112: 24,32%. Table legends are incomplete (add date of the study and location/ country). 95% CI should be added to the results.

Author’s Response 6: Thanks. Results section was revised based on comments.

Reviewer comment 7: Discussion: The authors made an attempt to emphasis the public health and animal health impact of the findings in aborted samples. Therefore, the need for surveillance programs to be implemented in Iran. However, one of the big pitfalls of the study is the sample size and convenience sampling, without knowing how representative are those samples of the herd prevalence, external validity, those claims are difficult to assess. One of the novel aspects of this study is that the authors used PCR comparing to most of studies used seroprevalence. How different are those studies: studied population, diagnostic testing ( Se, Sp), samples tested ( clinical versus surveillance), etc? The authors briefly described the prevalence in other studies/ countries but do not provide a justification for such numeric difference. This should be explored in the discussion section.

Author’s Response 7: Discussion section was revised based on comments.

Reviewer comment about small samples size and sampling in this study is correct. We added this point is a limitation in discussion section. 

All similar studies, which listed in the discussion section, used convenience sampling like our study. Samples of these studies belonged to investigation studies and clinical samples. Also, wWe mentioned that “One of the reasons for the high prevalence in our study was the using of a much more sensitive method (Real-time PCR) for detection of C. burnetii. Other reasons include different study areas and increased prevalence over time.”. 

 Reviewer comment 8: Pitfalls are not addressed in the manuscript.

Author’s Response 8: Limitations were added to discussion section.

---

## [Decision Letter · Decision Letter 1]

17 Feb 2021

PONE-D-20-32185R1

Molecular detection of Coxiella burnetii infection in aborted samples of domestic ruminants in Iran

PLOS ONE

Dear Dr. Esmaeili,

Thank you for submitting your manuscript to PLOS ONE. After careful consideration, we feel that it has merit but does not fully meet PLOS ONE’s publication criteria as it currently stands. Therefore, we invite you to submit a revised version of the manuscript that addresses the points raised during the review process.

The comments that were raised by the reviewers have not been adequately addressed in your revised manuscript. You still have to adequately address the issues of the English language, the qPCR used, the sample size and validity of the results as mentioned by the reviewers. Some of the clarifications that are given in the rebuttal letter are not included in the manuscript. Also make sure that the two versions of the manuscripts that are submitted are exactly the same. For example, the title for Table 1 is different between the two.  In addition, can you justify calculating percentages and 95% confidence intervals for small sample sizes. What value does it add to the interpretation of the results, if any? Please also give a description of your sampling sites in your materials and methods. 

We look forward to receiving your revised manuscript.

Kind regards,

Martin Chtolongo Simuunza, PhD

Academic Editor

PLOS ONE

Reviewers' comments:

Reviewer's Responses to Questions

**Comments to the Author**

1. If the authors have adequately addressed your comments raised in a previous round of review and you feel that this manuscript is now acceptable for publication, you may indicate that here to bypass the “Comments to the Author” section, enter your conflict of interest statement in the “Confidential to Editor” section, and submit your "Accept" recommendation.

Reviewer #1: (No Response)

Reviewer #2: All comments have been addressed

2. Is the manuscript technically sound, and do the data support the conclusions?

Reviewer #1: Yes

Reviewer #2: Yes

3. Has the statistical analysis been performed appropriately and rigorously? 

Reviewer #1: N/A

Reviewer #2: Yes

4. Have the authors made all data underlying the findings in their manuscript fully available?

Reviewer #1: Yes

Reviewer #2: Yes

5. Is the manuscript presented in an intelligible fashion and written in standard English?

Reviewer #1: Yes

Reviewer #2: Yes

6. Review Comments to the Author

Reviewer #1: The manuscript by Mobarez et al., describes a survey of 146 tissue samples from spontaneously-aborted cattle, sheep, and goats for the presence of Coxiella burnetii by RT-PCR, using IS1111 as a target. This is a revised version of the manuscript, but it still contains numerous items that need to be addressed.

Major issues-

1. The authors’ responses to Reviewer 1 comments 2 and 3 (i.e., information regarding how they established that ≤ 36 cycles was the RT-PCR Ct value for positivity, how many RT-PCR analyses were done per sample, replicates, etc., were not added to the revised manuscript. Please add this information to the materials and methods or at least cite a reference(s) so readers know how the PCR was done and verified.

2. The authors have transposed the headings on the actual S1 Table and S2 Table (they are correct on lines 176 and 177). They should read “S1 Table. Results of sheep abortion sampling” And “S2 Table. Results of cow abortion sampling”.

3. In certain instances the authors use average values rounded to the hundredth place, while in others they round to the tenth place. Please round up all figures to the tenth place throughout the paper for consistency. For example, line 20 should read 24.7% (not 24.66%).

4. Lines 21 and 107- The average value reported is not correct. 20 of 94 sheep samples is 21.3%, not 21.78% as shown.

5. Lines 100-102- The authors state that “94 samples…were from sheep abortions (S1 Table)”. However, if you look at S1 Table (after correcting for the heading transposition), there are 89 samples in column 3, not 94 as stated. Please clarify.

6. Lines 102-104- The authors state that “46 samples…were from cattle abortions (S2 Table)”. However, if you look at S2 Table (after correcting for the heading transposition), there are 48 samples in column 3, not 46 as stated. Please clarify.

Minor editorials (English, grammar, etc.)-

7. Line 17- should read “…to investigate the occurrence of C. burnetii…”

8. Line 18- should read “…from cattle, sheep and goats…” (cattle and sheep are plural in English)

9. Lines 19 and 86- should read “…targeting the IS1111…”

10. Line 25- should read “…paid to the role…”

11. Line 26- should read “…veterinary organizations.”

12. Indent for new paragraphs at lines 35, 51, 61, 77, 106, 112, 151 and 162

13. Line 33- should read “…16S rRNA …belongs to…”

14. Line 41- change “animals” to animals’ or animal

15. Line 47- should read “…febrile illness with…and is asymptomatic…”

16. Line 52- delete “the”

17. Line 53- should read “…are more likely to appear in small…”

18. Line 56- replace “dose” with “density”, as dose refers to load given to animal/human

19. Line 58- delete “transmitted” (redundant with previous line)

20. Lines 108 and 113- should read “The highest prevalence…”

21. Line 115- should read “…goat-aborted…” and “…four of the samples…”

22. Line 127- should read “…be seriously considered in cases…”

23. Line 128- should read “…to humans regarded.” and “…fever, yet it is still…”

24. Line 131- should read “…in the human health care…”

25. Line 139- replace “most” with “highest”

26. Line 140- should read “…Iran, wherein the …”

27. Line 144 and 158- replace “using” with “use”

28. Line 154- replace “with” with “in”

29. Line 155- should read “…countries, the prevalence…”

30. Line 162-163- should read “…to make prevalence comparisons.” Also replace “large” with “greater”

31. Line 164- should read “It is also recommended that non-convenience…”

32. Lines 166-167- should read “…fetuses, can inform the epidemiology…”

33. Line 192- do the authors mean large-animal veterinarians? Also- replace “helps” with “help”

34. Lines 143-144 and 157-158 are redundant. “One of the reasons…”

35. Table 1 legend shown with the actual table (lines 257-258) is not the same as that as shown on lines 118-119 of the text. It should read "...sheep and cattle abortion..." for consistency's sake.

Reviewer #2: Title: ‘Molecular detection of Coxiella burnetii in aborted samples of livestock in Iran’.

The authors addressed the main concerns of the reviewers, although, the discussion could be improved. The authors added most of the information requested by the reviewers, but there is some redundancy regarding the results, moreover comparison with other studies is very descriptive without a link or comparison between the present findings, and lacking on interpretation and potential reasons for similar (or not) findings (see examples below). I have some minor comments regarding the terminology/ grammar as the following:

Minor changes:

Line 18: remove ‘s , ‘ change to from cattle, sheep and goats’

Line 26 : ‘veterinary organization’, should be ‘veterinary organizations’?

Line 55: ‘of transmission in the environment’, change to contamination of the environment.

Line 60: ‘ in the transmission of the disease’ , change to ‘ in the transmission risk to humans’

Line 73 ‘cotyledons’, placenta cotyledons

Line 76: ‘…. Was done under a BSL2 biological safety cabinet’, maybe change to ‘…was done under biosafety level 2 (BSL2) laboratory settings’

Line 124: ‘ C. bunetti in domestic animal abortions’, change to ‘… in ruminant abortions’

Lines 127-129. I am not sure what the authors are trying to convey here. Please rephrase this sentence.

Lines 153-136: It seems it is missing a linking sentence. Suggestion: “Actually Q fever is one of the causes of endocarditis in humans, it was reported that about of 31 % ……”

Lines 140: remove ‘…., so that the….’ , change too ‘. Molecular prevalence….’

Lines 145: ‘…like our study..’, change to ‘ similar to our study’ . Here is one of the examples that I think the discussion is very simplistic. Although the authors added this sentence as response to the previous reviewer, they haven’t expanded on limitations of prevalence studies using convenience sampling, are those representative of the true prevalence?, are those studies comparable? etc.

Lines 149- 150. Another example of simplistic discussion, why we should give more attention to abortion in sheep due to C. burnetti.

Lines 167: ‘… better judgements..’’, change to ‘…inform better…’

Line 173 : ranchers or farmers?

7. PLOS authors have the option to publish the peer review history of their article (what does this mean?). If published, this will include your full peer review and any attached files.

Reviewer #1: No

Reviewer #2: No

---

## [Author Response · Author response to Decision Letter 1]

18 Feb 2021

Author’s Response to Reviewers Comment

Note: All changes showed in Revised Article with Changes Highlighted uploaded file and made by Track Change.

Reviewer 1#

Reviewer #1: The manuscript by Mobarez et al., describes a survey of 146 tissue samples from spontaneously-aborted cattle, sheep, and goats for the presence of Coxiella burnetii by RT-PCR, using IS1111 as a target. This is a revised version of the manuscript, but it still contains numerous items that need to be addressed.

Reviewer comment 1: The authors’ responses to Reviewer 1 comments 2 and 3 (i.e., information regarding how they established that ≤ 36 cycles was the RT-PCR Ct value for positivity, how many RT-PCR analyses were done per sample, replicates, etc., were not added to the revised manuscript. Please add this information to the materials and methods or at least cite a reference(s) so readers know how the PCR was done and verified.

Author’s Response 1: These points were added to manuscript. Also, refence of RT-PCR Ct value was added to manuscript.

Reviewer comment 2: The authors have transposed the headings on the actual S1 Table and S2 Table (they are correct on lines 176 and 177). They should read “S1 Table. Results of sheep abortion sampling” And “S2 Table. Results of cow abortion sampling.

Author’s Response 2: Thanks. Revised.

Reviewer comment 3: In certain instances, the authors use average values rounded to the hundredth place, while in others they round to the tenth place. Please round up all figures to the tenth place throughout the paper for consistency. For example, line 20 should read 24.7% (not 24.66%).

Author’s Response 3: All figures were Revised based on Reviewer comment.

Reviewer comment 4: Lines 21 and 107- The average value reported is not correct. 20 of 94 sheep samples is 21.3%, not 21.78% as shown.

Author’s Response 4: Thanks, revised.

Reviewer comment 5: Lines 100-102- The authors state that “94 samples…were from sheep abortions (S1 Table)”. However, if you look at S1 Table (after correcting for the heading transposition), there are 89 samples in column 3, not 94 as stated. Please clarify.

Author’s Response 5: 94 samples is correct. We forgot to add 5 samples of West Azerbaijan province in S1 Table. S1 Table was revised. Thanks for your attention. 

Reviewer comment 6: Lines 102-104- The authors state that “46 samples…were from cattle abortions (S2 Table)”. However, if you look at S2 Table (after correcting for the heading transposition), there are 48 samples in column 3, not 46 as stated. Please clarify.

Author’s Response 6: 46 sample is correct. S2 Table was revised.

Reviewer comment 7: Minor editorials (English, grammar, etc.) comments: 

7. Line 17- should read “…to investigate the occurrence of C. burnetii…”: Done.

8. Line 18- should read “…from cattle, sheep and goats…” (cattle and sheep are plural in English) : Done.

9. Lines 19 and 86- should read “…targeting the IS1111…”: Done.

10. Line 25- should read “…paid to the role…”: Done.

11. Line 26- should read “…veterinary organizations.” :Done.

12. Indent for new paragraphs at lines 35, 51, 61, 77, 106, 112, 151 and 162: Done.

13. Line 33- should read “…16S rRNA …belongs to…”: Done.

14. Line 41- change “animals” to animals’ or animal: Done.

15. Line 47- should read “…febrile illness with…and is asymptomatic…”: Done.

16. Line 52- delete “the”: Done.

17. Line 53- should read “…are more likely to appear in small…”: Done.

18. Line 56- replace “dose” with “density”, as dose refers to load given to animal/human: Done.

19. Line 58- delete “transmitted” (redundant with previous line): Done.

20. Lines 108 and 113- should read “The highest prevalence…”: Done.

21. Line 115- should read “…goat-aborted…” and “…four of the samples…”: Done.

22. Line 127- should read “…be seriously considered in cases…”: Done.

23. Line 128- should read “…to humans regarded.” and “…fever, yet it is still…”: Done.

24. Line 131- should read “…in the human health care…”: Done.

25. Line 139- replace “most” with “highest” :Done.

26. Line 140- should read “…Iran, wherein the …”: Done.

27. Line 144 and 158- replace “using” with “use” :Done.

28. Line 154- replace “with” with “in” :Done.

29. Line 155- should read “…countries, the prevalence…”: Done.

30. Line 162-163- should read “…to make prevalence comparisons.” Also replace “large” with “greater” :Done.

31. Line 164- should read “It is also recommended that non-convenience…”: Done. 

32. Lines 166-167- should read “…fetuses, can inform the epidemiology…”: Done. 

33. Line 192- do the authors mean large-animal veterinarians? Also- replace “helps” with “help : Done.

34. Lines 143-144 and 157-158 are redundant. “One of the reasons…”: Done.

35. Table 1 legend shown with the actual table (lines 257-258) is not the same as that as shown on lines 118-119 the text. It should read "...sheep and cattle abortion..." for consistency's sake. :Done.

Author’s Response 7: All done. Thank you very much for your valuable comments.

Reviewer 2#

Reviewer #2: Title: ‘Molecular detection of Coxiella burnetii in aborted samples of livestock in Iran’.

The authors addressed the main concerns of the reviewers, although, the discussion could be improved. The authors added most of the information requested by the reviewers, but there is some redundancy regarding the results, moreover comparison with other studies is very descriptive without a link or comparison between the present findings, and lacking on interpretation and potential reasons for similar (or not) findings (see examples below). I have some minor comments regarding the terminology/ grammar as the following.

Reviewer comment 1: Minor changes:

Line 18: remove ‘s , ‘ change to from cattle, sheep and goats’: Done.

Line 26 : ‘veterinary organization’, should be ‘veterinary organizations’? : Done.

Line 55:‘of transmission in the environment’, change to contamination of the environment.: Done.

Line 60: ‘ in the transmission of the disease’ , change to ‘ in the transmission risk to humans’: Done.

Line 73 ‘cotyledons’, placenta cotyledons: Done.

Line 76: ‘…. Was done under a BSL2 biological safety cabinet’, maybe change to ‘…was done under biosafety level 2 (BSL2) laboratory settings’ : Done.

Line 124: ‘ C. bunetti in domestic animal abortions’, change to ‘… in ruminant abortions’: Done.

Lines 127-129. I am not sure what the authors are trying to convey here. Please rephrase this sentence: Done.

Lines 153-136: It seems it is missing a linking sentence. Suggestion: “Actually Q fever is one of the causes of endocarditis in humans, it was reported that about of 31 % ……”: Done.

Lines 140: remove ‘…., so that the….’ , change too ‘. Molecular prevalence….’: Done.

Lines 145: ‘…like our study.’, change to ‘similar to our study’. Here is one of the examples that I think the discussion is very simplistic. Although the authors added this sentence as response to the previous reviewer, they haven’t expanded on limitations of prevalence studies using convenience sampling, are those representatives of the true prevalence? are those studies comparable? etc.: Done.

Lines 149- 150. Another example of simplistic discussion, why we should give more attention to abortion in sheep due to C. burnetti.: Done.

Lines 167: ‘… better judgements.’’, change to ‘…inform better…’ : Done.

Line 173: ranchers or farmers? Ranchers. 

Author’s Response 1: All done. Thank you very much for your valuable comments.

---

## [Decision Letter · Decision Letter 2]

29 Mar 2021

PONE-D-20-32185R2

Molecular detection of Coxiella burnetii infection in aborted samples of domestic ruminants in Iran

PLOS ONE

Dear Dr. Esmaeili,

Thank you for submitting your manuscript to PLOS ONE. After careful consideration, we feel that it has merit but does not fully meet PLOS ONE’s publication criteria as it currently stands. Therefore, we invite you to submit a revised version of the manuscript that addresses the points raised during the review process.

Please attend to these minor corrections before your paper can be accepted.

We look forward to receiving your revised manuscript.

Kind regards,

Martin Chtolongo Simuunza, PhD

Academic Editor

PLOS ONE

Journal Requirements:

Reviewers' comments:

Reviewer's Responses to Questions

**Comments to the Author**

1. If the authors have adequately addressed your comments raised in a previous round of review and you feel that this manuscript is now acceptable for publication, you may indicate that here to bypass the “Comments to the Author” section, enter your conflict of interest statement in the “Confidential to Editor” section, and submit your "Accept" recommendation.

Reviewer #1: (No Response)

Reviewer #2: All comments have been addressed

2. Is the manuscript technically sound, and do the data support the conclusions?

Reviewer #1: Yes

Reviewer #2: Yes

3. Has the statistical analysis been performed appropriately and rigorously? 

Reviewer #1: N/A

Reviewer #2: N/A

4. Have the authors made all data underlying the findings in their manuscript fully available?

Reviewer #1: Yes

Reviewer #2: Yes

5. Is the manuscript presented in an intelligible fashion and written in standard English?

Reviewer #1: Yes

Reviewer #2: Yes

6. Review Comments to the Author

Reviewer #1: The manuscript by Mobarez et al., is a second revision of the manuscript. The authors have addressed all my earlier concerns, but I have a few minor suggested changes for clarity and grammar's sake:

Line 33- replace “belonged” with “belongs” (Coxiella still belongs to this taxonomy.)

Line 35- indent paragraph for print editor

Line 54- should read “…main routes of contaminating the…”

Line 96- should read “…samples were tested…”

Line 97- should read “…positive samples were visualized by 2% agarose...

Lines 97-99- should read “…that had three positive PCR tests and also had the…. as true positives.”

Line 128- replace “Ira” with “Iran”

Line 129- replace “According to” with “Based on”

Line 138- should read “…humans, and it was…” (It is a run-on sentence in its current form.)

Line 139- delete “of”

Line 144-145- replace “north-east of Iran” with “northeastern Iran” and "south-eastern Iran" with "southeastern Iran"

Line 151- should read “…burnetii into milk…”

Line 166- replace “comparison” with “comparisons”

S1 and S2 Table headings- The headings should be replaced with those shown on line 179 of the text. (“Results of cow/sheep abortion sampling.”)

Reviewer #2: (No Response)

7. PLOS authors have the option to publish the peer review history of their article (what does this mean?). If published, this will include your full peer review and any attached files.

Reviewer #1: No

Reviewer #2: No

---

## [Author Response · Author response to Decision Letter 2]

29 Mar 2021

Author’s Response to Reviewers Comment

Note: All changes showed in Revised Article with Changes Highlighted uploaded file and made by Track Change.

Reviewer 1#

The manuscript by Mobarez et al., is a second revision of the manuscript. The authors have addressed all my earlier concerns, but I have a few minor suggested changes for clarity and grammar's sake:

Line 33- replace “belonged” with “belongs” (Coxiella still belongs to this taxonomy.)

Line 35- indent paragraph for print editor

Line 54- should read “…main routes of contaminating the…”

Line 96- should read “…samples were tested…”

Line 97- should read “…positive samples were visualized by 2% agarose...

Lines 97-99- should read “…that had three positive PCR tests and also had the….as true positives.”

Line 128- replace “Ira” with “Iran”

Line 129- replace “According to” with “Based on”

Line 138- should read “…humans, and it was…” (It is a run-on sentence in its current form.)

Line 139- delete “of”

Line 144-145- replace “north-east of Iran” with “northeastern Iran” and "south-eastern Iran" with "southeastern Iran"

Line 151- should read “…burnetii into milk…”

Line 166- replace “comparison” with “comparisons”

S1 and S2 Table headings- The headings should be replaced with those shown on line 179 of the text. (“Results of cow/sheep abortion sampling.”)

Author’s Response 1: Thanks. All done.

---

## [Editor Report · Decision Letter 3]

31 Mar 2021

Molecular detection of Coxiella burnetii infection in aborted samples of domestic ruminants in Iran

PONE-D-20-32185R3

Dear Dr. Esmaeili,

We’re pleased to inform you that your manuscript has been judged scientifically suitable for publication and will be formally accepted for publication once it meets all outstanding technical requirements.

Kind regards,

Martin Chtolongo Simuunza, PhD

Academic Editor

PLOS ONE
---

## [Editor Report · Acceptance letter]

5 Apr 2021

PONE-D-20-32185R3 

Molecular detection of *Coxiella burnetii* infection in aborted samples of domestic ruminants in Iran 

Dear Dr. Esmaeili:

I'm pleased to inform you that your manuscript has been deemed suitable for publication in PLOS ONE. Congratulations! Your manuscript is now with our production department. 

Kind regards, 

on behalf of

Dr. Martin Chtolongo Simuunza 

Academic Editor

PLOS ONE